# Primary Cutaneous B-Cell Lymphomas with Large Cell Morphology: A Practical Review

**DOI:** 10.3390/ijms24076204

**Published:** 2023-03-25

**Authors:** Andrea Ronchi, Paola Vitiello, Giuseppe D’Abbronzo, Stefano Caccavale, Giuseppe Argenziano, Antonello Sica, Roberto Alfano, Giovanni Savarese, Massimiliano Berretta, Immacolata Cozzolino, Renato Franco

**Affiliations:** 1Pathology Unit, Department of Mental Health and Physic and Preventive Medicine, University of Campania Luigi Vanvitelli, 80138 Naples, Italy; 2Dermatology Unit, Department of Mental and Physical Health and Preventive Medicine, University of Campania Luigi Vanvitelli, 80131 Naples, Italy; 3Oncology Unit, Department of Precision Medicine, University of Campania Luigi Vanvitelli, 80131 Naples, Italy; 4Department of Anesthesiology, Surgery and Emergency, University of Campania Luigi Vanvitelli, 80138 Naples, Italy; 5AMES, Centro Polidiagnostico Strumentale srl, 80013 Naples, Italy; 6Department of Clinical and Experimental Medicine, University of Messina, 98122 Messina, Italy

**Keywords:** primary cutaneous lymphoma, follicle center lymphoma, diffuse large B-cell lymphoma, leg type

## Abstract

Most primary cutaneous lymphomas consist of T-cell lymphomas or small cell lymphomas; however, the skin may also be affected by lymphomas with large cell morphology, as a primary or secondary localization. A minority of cases consist of primary cutaneous B-cell lymphomas (PCBCLs). PCBCLs are a heterogeneous group of rare neoplasms with an overlapping morphological and immunohistochemical picture of the different subtypes. Nevertheless, differential diagnosis in the setting of this group of neoplasms is mandatory to identify the correct therapy and prognosis, but it may be challenging since, due to the rarity of these neoplasms, they may not always be familiar to pathologists. Indeed, immunohistochemistry may not be enough to distinguish the different histotypes, which overlap in immunohistochemical features. Furthermore, the ever-increasing knowledge of the molecular features of systemic B-cell lymphomas, such as gene rearrangements with clinical significance, has led in recent years to further investigation into the molecular landscape of PCBCLs with large cell morphology. This work aimed to provide a practical diagnostic guide for pathologists dealing with primary cutaneous large B-cell lymphomas.

## 1. Introduction

Most primary cutaneous lymphomas are constituted by small cells. T-cell lymphomas, deriving from mycosis fungoides, are the most common. However, lymphomas with large cell morphology may involve the skin, as a primary localization, or secondary involvement. Primary cutaneous large cell lymphomas include mainly T-cell lymphomas, such as large cell transformation of mycosis fungoides, anaplastic lymphoma, lymphomatoid papulosis type C, some cases of aggressive cytotoxic cutaneous lymphoma and peripheral T-cell lymphoma, NOS [1,2]. Primary cutaneous B-cell lymphomas (PCBCLs) with large cell morphology are a heterogeneous group of rare neoplasms, constituted histologically by various proportions of large cells with the morphological features of centroblasts and/or immunoblasts, with the interposition of other lymphoid and non-lymphoid cells (centrocytes, follicular dendritic cells, small reactive lymphocytes) (Figure 1).

The classification of PCBCLs with large cell morphology (which will remain unaltered in the upcoming edition of WHO classification [3]) includes primary cutaneous diffuse large B-cell lymphoma, leg type (PCDLBCL-LT), primary cutaneous follicle center lymphoma (PCFCL) and primary cutaneous diffuse large B-cell lymphoma, other (PCDLBCL-O) [4]. PCBCLs are lymphomas arising in the skin, without clinical and instrumental evidence of systemic disease at the time of the diagnosis, constituted by at least 25–30% of large cells (cells at least 4 times the size of a small lymphocyte). The diagnosis of PCBCLs cannot be made only based on histological findings, since it requires the demonstration of skin-limited disease at the time of the diagnosis, by total-body instrumental staging [5]. The differential diagnosis between this group of neoplasms may be challenging due to the rarity of these neoplasms and the overlapping morphological and immunohistochemical features of the subtypes. Immunohistochemistry is always needed and may be helpful, but it may not be enough to distinguish the different histotypes, which overlap in immunohistochemical features, and the differential diagnosis is often based on the combination of morphological and immunohistochemical details. Nevertheless, the correct diagnosis is mandatory, since prognosis and therapy are significantly different, depending largely on the histological diagnosis [6]. Moreover, improved knowledge of the molecular features of systemic B-cell lymphomas, including the gene rearrangements with clinical significance, has led in recent years to further investigation into the molecular landscape of PCBCLs with large cell morphology.

This review summarizes current knowledge on the clinical, morphological, immunohistochemical, and molecular findings of PCBCLs with large cell morphology, functioning as a practical guide for the diagnosis in the clinical setting. A literature search was performed in PubMed and Web of Science for studies about the clinical, pathological, and molecular findings of PCBCLs with large cell morphology, using the following search terms: primary cutaneous B-cell lymphoma, primary cutaneous diffuse large B-cell lymphoma leg type, primary cutaneous follicle center lymphoma, primary cutaneous diffuse large B-cell lymphoma other. Only English publications were included.

## 2. Primary Cutaneous Diffuse Large B-Cell Lymphoma, Leg Type (PCDLBCL-LT)

### 2.1. Clinical Findings and Behavior

PCDLBCL-LT is an infrequent neoplasm, representing 1–4% of all cutaneous lymphomas and about 20% of all PCBCLs [7]. Elderly women are affected more often than males. Lower extremities are the most common sites of onset of PCDLBCL-LT, but it may arise in other cutaneous sites in about 10–15% of cases. Localizations to head or neck are rare. Patients often show multiple lesions, represented by reddish plaques or nodules, which may be ulcerated, but a single lesion may be possible. PCDLBCL-LT has an intermediate prognosis, with 5-year disease-specific survival of about 60% and an overall survival of about 50% [8]. Progression-free survival is about 41.8 months in patients treated with chemotherapy and radiotherapy [9]. Cutaneous dissemination and relapses are common, and systemic dissemination develops in about 17–47% of cases, mainly to lymph nodes. Adverse prognostic features include multiple lesions, ulceration, CDKN2A inactivation, and MYC rearrangement [8,9,10,11]. The prognostic role of MYD88 mutation and bcl2 expression is debated. Front-line therapy for PCDLBCL-LT includes R-CHOP with or without involved-site radiation therapy [9]. Some data suggest that immune checkpoint inhibitors may have a role in the treatment of relapsed/refractory cases [9].

### 2.2. Histological Findings

PCDLBCL-LT is defined as a primary cutaneous lymphoma composed exclusively of centroblasts and immunoblasts, most commonly arising in the leg [4]. As suggested by the definition, the most important diagnostic clue of this lymphoma is its cellular composition as assessed by histological examination, with the neoplastic population constituted only by centroblasts and immunoblasts (Figure 2).

The lymphoid proliferation is organized in diffuse sheets, occupying diffusely the dermis, with variable involvement of the sub-cutaneous fat. Upon low power field observation, the visual impression is that of a monotonous cell population, being constituted by only two cytotypes. Importantly, there are only a few small lymphocytes (reactive T-cells) in the background, often confined to perivascular areas, and follicular dendritic cells are absent. When the lesion is ulcerated, an inflammatory component of granulocytes and plasma cells may be present at the bottom of the ulcer.

Immunohistochemically, PCDLBCL-LT expresses all the pan-B markers, such as CD20, CD19, and Pax5. Pan-T markers may be helpful to quantify the reactive T-cells in the background, and CD21 and CD23 may help to confirm the absence of follicular dendritic cells. The prototype immunohistochemical profile of PCDLBCL-LT includes positivity to MUM1, bcl6 (often slight intensity), bcl2, FOXP1, and IgM, and negativity for CD10, CD30, and CD5. However, CD10 may be positive, often with slight intensity. On the other hand, bcl6, MUM1, bcl2, FOXP1, and IgM may be negative. Although PCDLBCL-LT is considered an activated B-cell lymphoma and it usually shows a non-germinal center (GC) phenotype according to Hans’s algorithm, a GC-phenotype is possible and not infrequently observed. Bcl2 is expressed in most cases (from 94–100% in the largest series), and its positivity is helpful for the diagnosis of PCDLBCL-LT, since PCGCL is usually (but not always) negative [10,11,12]. The proliferation index (Ki67) is high—more than 50%. C-myc is expressed in 67–83% of cases [9,10,11]. The main pathological findings for diagnostic purpose are summarized in Table 1. Immunohistochemical findings are listed in Table 2. A similar percentage of cases (69–83%) co-express bcl2 and c-myc (dual expressors) [9,10,11,12]. Although the prognostic significance of dual expressor status is still not entirely clear, it displayed a significantly worse overall survival and specific survival in the study of Menguy et al. [11].

### 2.3. Molecular Findings

The most common molecular alteration found in PCDLBCL-LT is related to the B-cell receptor pathway activation and dysregulation of the NF-kB signaling pathway, which promotes cell survival, proliferation, and the inhibition of apoptosis in lymphoid cells [13]. Activating mutations of MYD88 (mainly MYD88 L265P) and CD79B (mainly ITAM domain) are the most common hot spot mutations in PCDLBCL-LT and are both useful for diagnostic purposes. Activating mutations of the coiled-coil domain of CARD11 and heterozygous deletions of A20 are also common [14]. PCDLBCL-LT harbors the molecular signature of “activated B-cell-like” lymphomas showing a terminal B-cell differentiation blockage and resembles primary large B-cell lymphoma of immune-privileged sites, such as in the central nervous system and testis lymphomas. Mareschal et al. analyzed the molecular profile of 20 cases of PCDLBCL-LT, showing a very restricted set of highly recurrent mutations, including MYD88 (75% of cases), PIM1 (70% of cases), CD79B (40% of cases), and others (TBL1XR1, MYC, CREBBP, IRF4, HIST1H1E). Moreover, the authors reported some common genetic losses involving CDKN2A/2B, TNFAIP3/A20, PRDM1, TCF3, and CIITA [15]. The inactivation of CDKN2A (by either deletion or promoter hypermethylation) has a prognostic role in PCDLBCL-LT, and cases harboring homozygous deletion have poorer prognosis than cases harboring heterozygous deletion [16]. PCDLBCL-LT may harbor translocations of MYC, BCL6, and BCL2, but BCL2 translocation seems to be rare. In the largest series, translocations of MYC, BCL6, and BCL2 were demonstrated in 5–44% of cases, 4–29% of cases, and 0–12% of cases, respectively [10,11,12]. In the study of Schrader et al., 14 out of 44 (32%) cases showed MYC rearrangement, and 2 cases (4%) showed BCL6 rearrangement [10]. Double and triple hit status is possible but infrequent (19% and 6% of cases, respectively) [12]. The prognostic role of these cytogenetic alterations in PCDLBCL-LT is unclear. In the study of Schrader et al., MYC translocation correlated with disease-specific survival and disease-free survival, but not with overall survival [10]. Recent data suggest that immune escaping may be an important ontogenetic mechanism in PCDLBCL-LT, which harbors recurrent alterations in immune-evasion genes, such as PDL1/PDL2 translocations, leading to the overexpression of PD-L1 or PD-L2 proteins [13,17].

## 3. Primary Cutaneous Follicle Centre Lymphoma (PCFCL)

### 3.1. Clinical Findings and Behavior

PCFCL mainly affects middle-aged adults. Lesions are typically located on the head and neck or the upper trunk and are constituted by solitary or grouped papules, plaques, and/or tumors. On approximately 5% of cases, the legs are involved. Furthermore, 15% of patients present multifocal skin lesions. Ulceration may occur. PCFCL is an indolent disease, with a good prognosis. Five-year disease-specific survival is over 95%, and systemic spread is rare. Cutaneous relapses may occur (about 30% of cases) and tend to occur at the site of initial presentation. Cases located to the leg may have a more aggressive behavior [18].

### 3.2. Histological Findings

PCFCL is defined as a primary cutaneous lymphoma composed of centrocytes and a variable number of centroblasts, with a follicular, follicular and diffuse, or diffuse pattern of growth [4]. The cellular composition (centrocytes and centroblasts), which defines the neoplasms, is the most important clue for the diagnosis (Figure 3).

Since the centroblasts may be present in PCDLBCL-LT, centrocytes are mandatory for the diagnosis of PCFCL, and they are the most important clue in the differential diagnosis with other PCBCLs with large cell morphology. The presence of centrocytes is mandatory for the diagnosis of PCFCL, and lymphomas including centroblasts are excluded from this category. Overall, PCFCL is characterized by a dermal/subcutaneous infiltration by admixed centrocytes and centroblasts, often with an evident grenz zone. Variable numbers of reactive T-cells are present in the background. Although intermediate types do exist, PCFCL includes two morphological types: follicular type and diffuse type. The former is organized in follicles, sharing most of its diagnostical clues with the classic systemic follicular lymphoma. It must be differentiated by florid follicular hyperplasia (pseudo-lymphoma), marginal zone lymphoma, and systemic follicular lymphoma. The latter shows a diffuse growth pattern. It must be differentiated mainly by other lymphomas with large cell morphology, first PCDLBCL-LT and PCDLBCL-O. In the follicular type, the follicles are the main source of diagnostic features. Indeed, follicles are homomorphous in both diameter and cellular composition (often with the prevalence of centroblasts), follicular histiocytes are absent, mitoses are few and the proliferation index is relatively low, no signs of follicular polarization are apparent, the mantle zone is attenuated or absent (more evident by IgM IHC), and follicle center cells are present out of the dendritic follicular cells meshwork (more evident by CD21/CD23 and bcl6 IHC) (Figure 4) (Table 1). In the diffuse type, the neoplastic population is constituted by a monotonous population of large centrocytes, some of which may have a multilobated appearance, and centroblasts. There is no evidence of follicles, and follicular dendritic cell meshworks are absent (Table 1).

Immunohistochemically, PCFCL expresses all the pan-B markers, such as CD20, CD19, and Pax5. The prototype immunohistochemical profile of PCFCL includes positivity for bcl6, partial or absent expression of CD10, and negativity for MUM1, CD5, bcl2, and FOXP1; CD3 highlights a variable number of reactive T-cells; CD21 and CD23 highlight a disrupted meshwork of follicular dendritic cells in follicular type. The proliferation index (Ki67) is more often <50%. However, bcl6 and CD10 may both be negative, while MUM1, FOXP1, and bcl2 may be positive, and the proliferation index (Ki67) may be high (>50%) (Table 3). PCFCL diffuse type is characterized by CD10 negativity and high Ki67, while follicular dendritic cell meshworks are absent, BCL2, MUM1, and FOXP1 are usually negative, and CD5 and CD43 are always negative (Figure 5).

C-myc expression is not rare, as it is reported in up to 48% of cases [11]. Although the main diagnostic clues of follicular lymphoma bcl2 expression and BCL2 rearrangement–are often lacking in PCFCL, they may be seen in PCFCL. Bcl2 expression is seen in a significant proportion of cases (up to 38%) and is correlated with a higher risk of cutaneous relapses [11,19,20,21]. Up to 30% of cases co-express c-myc and bcl2 [11]. A low proliferation index (Ki67 < 30%) has been correlated with systemic spread [20].

### 3.3. Molecular Findings

The molecular landscape of PCFCL is not clearly defined, and actually there is no molecular marker that is able to distinguish this lymphoma from systemic follicular lymphoma or to predict future systemic involvement in PCFCL. However, BCL2 rearrangement is certainly common in systemic follicular lymphoma and rare in PCFCL [22,23]. BCL2, BCL6, and MYC rearrangements are rare in PCFCL. Indeed, in the series of Menguy et al., BCL6 and MYC rearrangements were found in 1 out of 21 (5%) cases [11]. However, BCL2 rearrangement has been variably reported in different series and may be found in up to 30% of cases [23,24,25,26,27]. Although the prognostic role of these molecular alterations is not entirely known, they seem not to affect overall survival or specific survival [11]. Zhou et al. recently indagated the molecular findings of skin-restricted PCFCL and the cutaneous involvement of systemic follicular lymphoma (SFL) with concurrent or future systemic involvement [20]. BLC2 rearrangement was found in 17% and 100% of PCFCL and SFL, respectively. By whole-exome sequencing, the authors demonstrated mutations in genes associated with chromatin remodeling in SFL, including CREBBP, KMT2D, and EZH2. On the other hand, PCFCL was characterized by a more heterogeneous molecular landscape, including mutations of TNFRSF14, MYC, JAK3, KRAS, FOXO1, CARD11, RHOA, TET2, SOCS1, and B2M [20].

## 4. Primary Cutaneous Diffuse Large B-Cell Lymphoma-Other (PCDLBCL-O): Primary Cutaneous Diffuse Large B-Cell Lymphomas Not Otherwise Specified (PCDLBCL, NOS)

PCDLBCL-O is not a specific pathological entity, but a heterogeneous group of B-cells neoplasms characterized by large cell morphology, which do not meet the diagnostic criteria for PCDLBCL-LT or PCFCL. This “umbrella” category includes rare specific sub-types, such as intravascular large B-cell lymphoma and cutaneous plasmablastic lymphoma and diffuse large B-cell lymphomas occurring primarily in the skin (primary cutaneous diffuse large B-cell lymphomas not otherwise specified, PCDLBCL, NOS).

### 4.1. Clinical Findings and Behavior

PCDLBCL, NOS is a poorly defined entity, and it is not entirely known whether it is an independent entity with respect to systemic DLBCL. PCDLBCL, NOS affects adults and elderly patients, with a median age of 70 years [18]. Although PCDLBCL, NOS is located most often on the trunk or head/neck, the anatomic distribution of the neoplasm is wide, and it arises on the leg in a variable percentage of cases. In the series of Kodama et al., five out of nine (55.6%) cases were located on the leg [18]. PCDLBCL, NOS has an intermediate prognosis, with a 5-year overall survival and disease-specific survival of about 50% [18].

### 4.2. Histological Findings

PCDLBCL, NOS is a diagnosis by exclusion and includes cases with large cell morphology not fulfilling diagnostic criteria for PCDLBCL-LT or PCFCL [28,29]. Histologically, PCDLBCL, NOS is constituted by a diffuse lymphoid population mainly of large cells, with variable morphology, including centroblasts, immunoblasts, and medium-sized centrocytoid cells. The lymphoid population is located in the dermis, but it extends to the hypodermis in about 60% of cases. The arrangement of the neoplasm may be nodular or diffuse, but it is more often mixed nodular and diffuse. Although the reactive T-cell population in the background is variable and may be scant in some cases, it is usually moderate or intense. A meshwork of follicular dendritic cells may be present. Similar to systemic DLBCL, immunohistochemistry is variable in the case of PCDLBCL, NOS, and a prototype immunophenotype is not defined. CD10, bcl6, and MUM1 are expressed in about 30%, 80%, and 25% of cases, respectively [12]. A “non-GC” phenotype according to Hans algorithm is more frequently observed [30,31]. Moreover, bcl2 and c-myc are expressed in about 65% and 35% of cases, respectively; however, the co-expression of these two markers is significantly more common in PCDLBCL-LT than in PCDLBCL, NOS [12]. The proliferation index (Ki67) is variable, and the mean value is about 40%. BCL2 and MYC translocation may be present in PCDLBCL, NOS, but the double-hit status is significantly more common in PCDLBCL-LT.

### 4.3. Molecular Findings

Molecular findings of PCDLBCL, NOS are not entirely known, and it is debatable whether the molecular landscape of this neoplasm is significantly different from systemic DLBCL. Weissinger Se et al. have recently indagated the molecular alteration in a series of primary extranodal lymphomas, including 16 PCDBCL, NOS [32]. The most common mutated genes in PCDLBCL, NOS were MYD88 (50%), CD79B (37%), CARD11 (6%), and BTK (6%). MYD88 and CD79B mutations are involved in B-cell receptor (BCR) activation and are significantly associated with a non-GC phenotype in accordance with Hans algorithm. In addition, MYD88 mutation seems to be significantly associated with a worse prognosis [32,33]. Alterations of PDL1/2 locus (9p24.1) are present in a variable number of cases, mainly as relative gain. Relative loss and polysomy of 9p24.1 have also been reported [32].

## 5. Conclusions

Diagnosis of PCBCLs with large cell morphology is mandatory for the correct management of the patients, as this group of neoplasms includes both indolent and aggressive subtypes and necessitates different therapies. The distinction between PCBCLs and secondary localization to the skin from a systemic lymphoma always needs clinical and instrumental evidence. The immunohistochemical and molecular features of PCBCLs are not entirely specific for each subtype, and a comprehensive evaluation of all clinical, histological, immunohistochemical, and molecular findings is needed. In some cases, morphological features are still the fundamental basis of the diagnosis, and a “prototype” immunophenotype is a helpful finding (Figure 6). More data are needed to establish whether PCDLBCL, NOS should be classified as an independent entity. Although T-cell lymphomas are the most common primary cutaneous lymphomas, dermatologists and pathologists should be familiar with PCBCLs with large cell morphology.

## Figures and Tables

**Figure 1 ijms-24-06204-f001:**
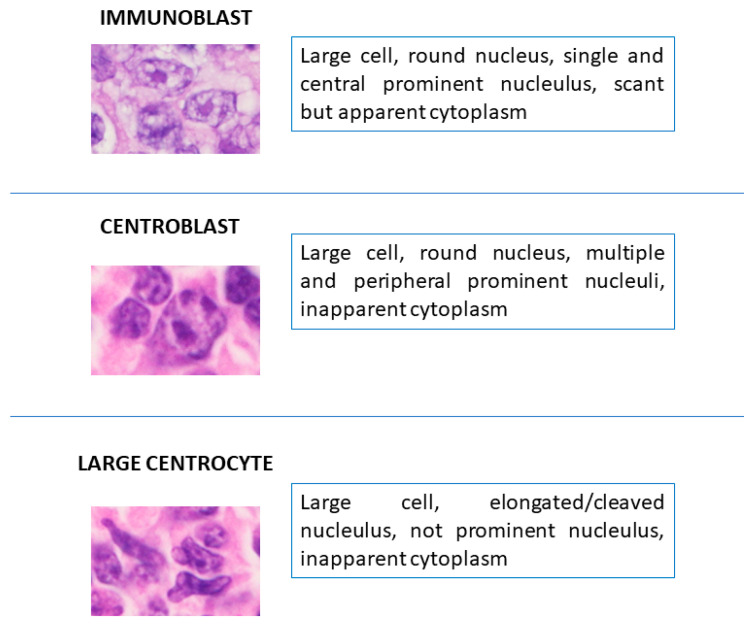
Most common cytotypes in primary cutaneous B-cell lymphomas with large cell morphology (hematoxylin and eosin stain, original magnification 650×).

**Figure 2 ijms-24-06204-f002:**
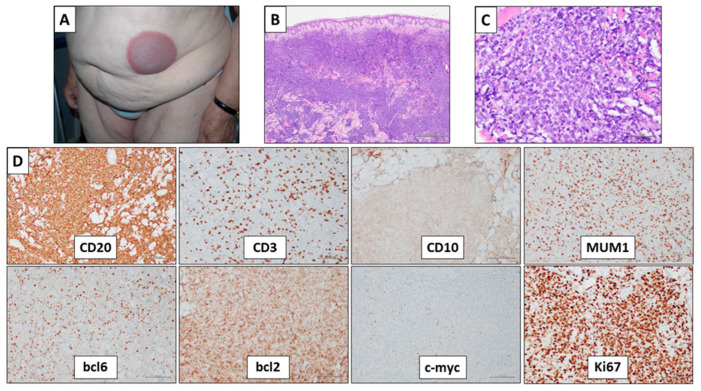
A case of primary cutaneous diffuse large B-cell lymphoma, leg type (PCDLBCL-LT). (**A**) A 76-year-old woman presenting a left-sided flank nodular lesion. The patients also showed a similar nodular lesion on the right scapular site and a poorly defined plaque lesion on the left leg. (**B**) Microscopic examination showing a diffuse lymphoid population filling the intermediate and deep dermis and sparing the papillary dermis (hematoxylin and eosin stain, original magnification 40×). (**C**) The neoplastic population includes large cells, with the morphological details of centroblasts and immunoblasts. Occasional small lymphocytes are present in the background (hematoxylin and eosin stain, original magnification 400×). (**D**) Immunohistochemistry showed positivity for CD20, MUM1, bcl6, and bcl2; slight positivity for CD10; negativity for CD3 (confirming the small number of reactive small T-cells) and c-myc. Proliferation index (Ki67) was about 80% (immunohistochemical stains, original magnification 200×).

**Figure 3 ijms-24-06204-f003:**
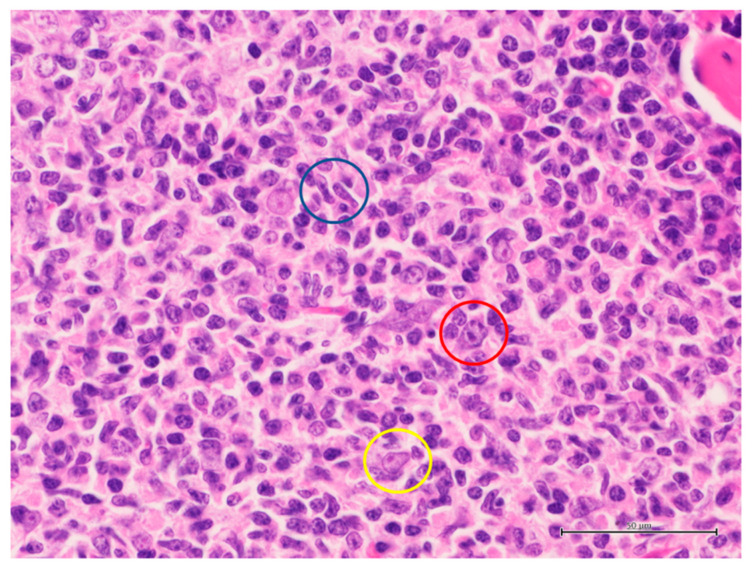
Cytotypes in PCFCL. Blue circle: large centrocytes, characterized by stretched irregular nuclei; red circle: centroblast, characterized by a roundish nucleolus with single prominent nucleolus; yellow circle: follicular dendritic cell, characterized by a poorly defined cytoplasm and an oval nucleolus with poorly evident nucleolus (hematoxylin and eosin stain, original magnification 400×).

**Figure 4 ijms-24-06204-f004:**
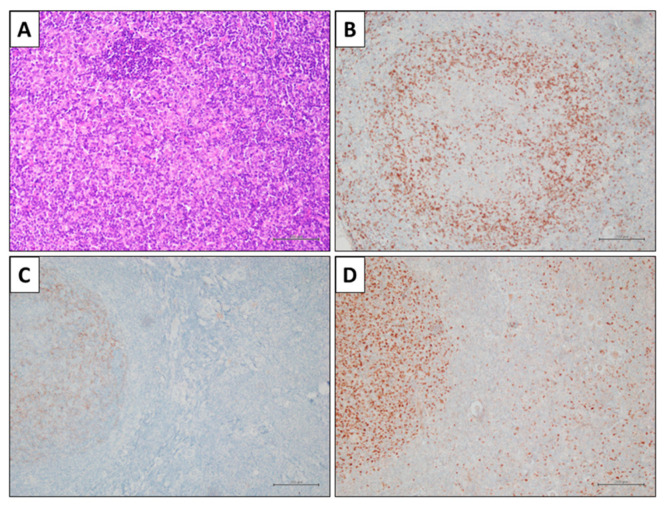
Diagnostic utility of follicle center (**A**) in PCFCL. (**B**) IgD immunohistochemistry showing a disrupted mantel layer. Bcl6-positive follicle center cells (**D**) are present outside of a CD21-positive dendritic cell meshwork (**C**) (immunohistochemical stains, original magnification 100×).

**Figure 5 ijms-24-06204-f005:**
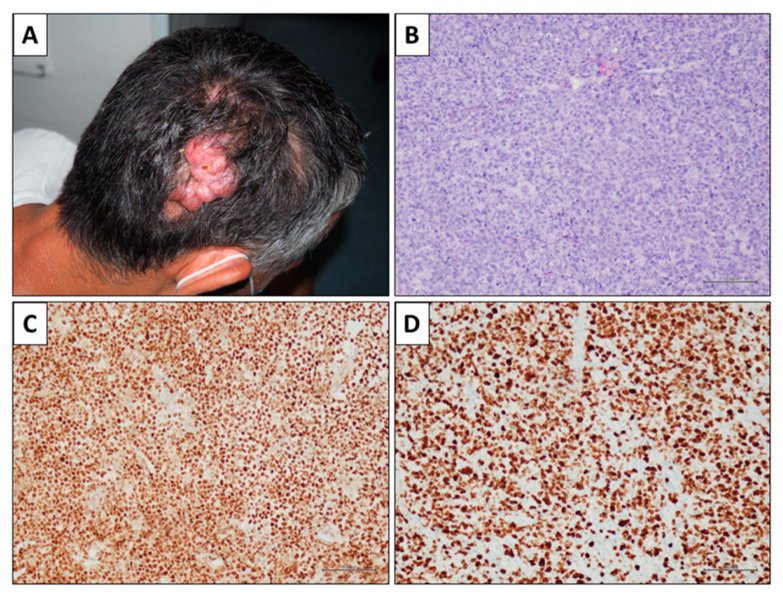
A case of primary cutaneous follicle center lymphoma (PCFCL). (**A**) A 62-year-old man presenting a nodular lesion on the occipital skin. (**B**) Microscopic examination showing a diffuse lymphoid population with large cell morphology (hematoxylin and eosin stain, original magnification 200×). (**C**) The neoplastic population expressed bcl6 and the proliferation index (Ki67) was high (**D**) (immunohistochemical stains, original magnification 200×).

**Figure 6 ijms-24-06204-f006:**
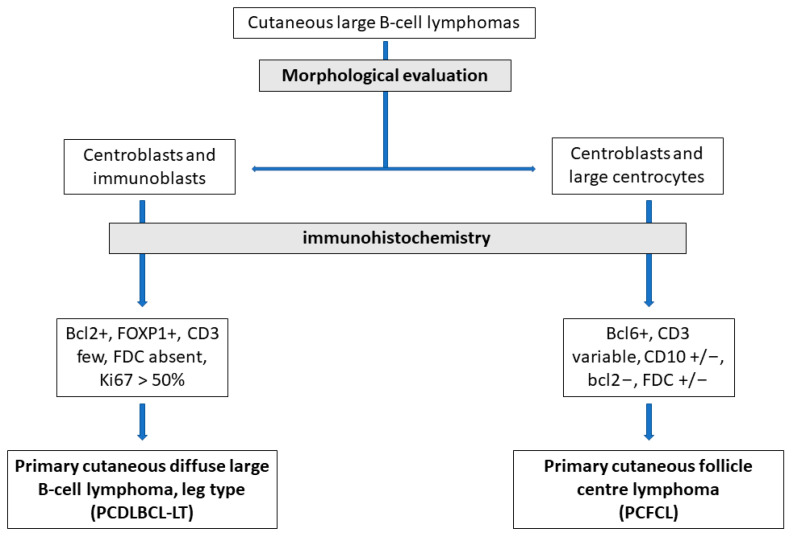
Diagnostic algorithm for the differential diagnosis between PCDLBCL-LT and PCFCL.

**Table 1 ijms-24-06204-t001:** Main diagnostic findings of primary cutaneous large B-cell lymphomas.

Histotype	Pathological Key Features	Helpful Finding	Diagnostic Molecular Tests
Primary cutaneous diffuse large B-cell lymphoma, leg type (PCDLBCL-LT)	- Neoplastic cells with features of immunoblasts and centroblasts - Few T-cells in the background - No follicular dendritic cell meshworks - No centrocytes	- Localization to the leg - Non-GC type immunophenotype - Prototype immunophenotype - Bcl2 positivity	- MYD88 and CD79B mutations - IGH, MYC, and BCL6 translocation - BCL2 translocation exceptional
Primary cutaneous follicle center lymphoma (PCFCL), follicular type	- Monomorphic follicles: many centroblasts, few/no histocytes, few/no mitoses, no polarization - Presence of follicular dendritic cell meshworks - Attenuated/absent mantle zone - Presence of follicle center cells out of follicular dendritic cell meshwork	- Localization to head and neck or upper trunk - Prototype immunophenotype - Bcl2 negativity - Absent Bcl2 translocation	- IGH rearrangement with somatic hypermutation - BCL2 rearrangements - Deletion of chromosome 14q32.33 - Mutations in MYD88 and inactivation of CDKN2A and CDKN2B by deletion (9p21.3) or their promotor hypermethylation is not or only rarely found
Primary cutaneous follicle center lymphoma (PCFCL), diffuse type	- Monomorphic population of large centrocytes and centroblasts	- Localization to head and neck or upper trunk - Many reactive T-cells - Prototype immunophenotype	

**Table 2 ijms-24-06204-t002:** Primary cutaneous diffuse large B-cell lymphoma, leg type (PCDLBCL-LT) immunohistochemistry.

	CD20	CD3	FDC	CD10	Bcl6	MUM1	FOXP1	CD5	Bcl2	Ki67
Prototype	+	few	absent	−	+ (dim)	+	+	−	+	>50%
Other				+ (dim)	−	−	−		−	

Abbreviations: FDC: follicular dendritic cells (stained by CD21 and CD23); +: positive; −: negative; dim: diminished (intensity).

**Table 3 ijms-24-06204-t003:** Primary cutaneous follicle center lymphoma (PCFCL) immunohistochemistry (IHC).

	CD20	CD3	FDC	CD10	Bcl6	MUM1	FOXP1	CD5	Bcl2	Ki67
Prototype	+	variable	+/−	+/−	+	−	−	−	−	<50%
Other			+	−	−	+	+		+	>50%

Abbreviations: FDC: follicular dendritic cells (stained by CD21 and CD23); +: positive; −: negative.

## Data Availability

Not applicable.

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
