# Peer review of "Primary Cutaneous B-Cell Lymphomas with Large Cell Morphology: A Practical Review"

_ijms, 2023, doi:10.3390/ijms24076204_

Round 1
Reviewer 1 Report
Dear authors, thank you for this helpful and well-conceptualized review of these rare large B-cell lymphoma subtypes. I have a few suggestions for improving the readability of the text. I would suggest adding one composite table including all 3 subtypes discussed instead of 3 separate tables. This will make it visually more appealing and have differences at a glance. A section on molecular subtyping and differences between the 3 subtypes can be added as a section in this table. There are some language changes (they are minor and don't reflect the intellectual aspect of this paper) I would advise for improving readability- for example: Instead of "Lower extremities of elderly woman are the most common sites of onset"- PCDLBCL-LT most commonly affects elderly women with an onset in the lower extremities". A few sentences require modification- for example, in the section on PCFCL immunohistochemistry- "partial or absent negativity for CD10" uses a double negative.
Another thing that will further enhance this manuscript is a diagnostic algorithm for accurate classification of cutaneous B-cell neoplasms and perhaps adding a sentence or two on whether there are data on novel targeted agents in this setting in combination with chemoimmunotherapy.
Thank you!
Author Response
Reviewer #1:
Dear authors, thank you for this helpful and well-conceptualized review of these rare large B-cell lymphoma subtypes. I have a few suggestions for improving the readability of the text. I would suggest adding one composite table including all 3 subtypes discussed instead of 3 separate tables. This will make it visually more appealing and have differences at a glance. A section on molecular subtyping and differences between the 3 subtypes can be added as a section in this table.
AA: Thank you for your interest in our work, and for your suggestion. The three tables about diagnostic findings of the different subtypes have been merged into a single table (table 1). A section on molecular features has been added to this table.
There are some language changes (they are minor and don't reflect the intellectual aspect of this paper) I would advise for improving readability- for example: Instead of "Lower extremities of elderly woman are the most common sites of onset"- PCDLBCL-LT most commonly affects elderly women with an onset in the lower extremities".
AA: The sentence has been modified based on the suggestions of the reviewer 2 too.
A few sentences require modification- for example, in the section on PCFCL immunohistochemistry- "partial or absent negativity for CD10" uses a double negative.
AA: The sentence has been modified. Moreover, we modified several sentences, according to the suggestions of the Reviewer 2 too.
Another thing that will further enhance this manuscript is a diagnostic algorithm for accurate classification of cutaneous B-cell neoplasms and perhaps adding a sentence or two on whether there are data on novel targeted agents in this setting in combination with chemoimmunotherapy.
AA: We added a figure showing a diagnostic algorithm including the two main histotypes. We did not include DLBCL-O, as this is an exclusive diagnosis. Furthermore, we added two sentences about systemic therapy for leg-type lymphoma (lines 94-96).

Reviewer 2 Report
Review ijms-2282588
In their manuscript “Primary cutaneous B-cell lymphomas with large cell morphology: a practical review.” Ronchi et al. review literature about rare primary b-cell lymphoma of the skin. Expanding upon the WHO-HAEM4R classification, the authors give interesting insight about the current molecular understanding of those entities and furthermore provide an argumentative baseline in favour or against defining PCBCL-NOS as separate entity.
Some formal/minor editing and aspects as well expansion upon methodical conception (see below) is needed before this review should be considered for publication.
Formal
- Figures in the manuscript feature cincial findings and tissue samples. Please state if those are either original (unpuplished) data (if so, is the ethical statemanet truley not appicable) or are those cited from an other publication (if so, citations are, of course, mandatory).
- Et al. is usually abbreviated with a point. Please check this in the manuscript.
- On several occasions a comma is used incorrectly before and. Please adjust this throughout the manuscript.
Major
Currently the manuscript is lacking important information about its methodology. The authors should elaborate on which merit articles were deemd suitable for inclusion into the review. Especially, was a (semi-)structured literature search used and which databeases were included? Depending on the lenght of the methodology section this could either be presented as seperate chapter or a paragraph at the end of the introduction.
Minor
- Line 19 (Abstract): The term "of the latter" is confusing in this context, mentioning secondary b-cell lymphoma of the skin just previously. I recommend not including it here and starting the sentence with: “A minority of cases (…)”
- Lines 25 & 59: Please replace "even overlapping (…)" with which are overlapping (…) or featuring overlapping to increase readability.
- Lines 34/35 (Introduction): Please rephrase here as two sentences, like: Most primary cutaneous lymphomas include small cells. T-cell lymphomas, deriving from mycosis fungoides, are the most common.
- Lines 48-50 (Introduction): Considering the upcoming WHO-HAEM5 which makes no alterations to PCBCL subtypes, I recommend outlining this and providing suitable citation [1]. Like: "Classification of PCBCLs with large cell morphology (which will remain unaltered in the upcoming edition of WHO classification) (…)"
- Lines 70 & 72 (PCDLBCL-LT): Occurrence in elderly women is stated somewhat redundantly. I suggest deleting it in the first sentence and rephrasing the latter one to: "Elderly Women are affected more often than males."
- Line 72 (PCDLBCL-LT): Please rephrase to: “Localization to head or neck are rare.”
- Line 73 (PCDLBCL-LT): Statement of the usual clinical presentation is used redundant. Please delete either "clinically" or "at the time of clinical observation" at the end of this sentence.
- Lines 75/76, 174/175, 272/273: Please elaborate upon the type of survival (overall, progression free, etc) if this is stated in the cited papers.
- Lines 76-79 (PCDLBCL-LT): Please provide suitable citation on prognostic factors of PCDLBCL-LT.
- Line 112 (PCDLBCL-LT): Germinal centre should be written out before abbreviating it the first time.
- Lines 115/116 (PCDLBCL-LT): Considering it is firstly stated in the text, please add: “Proliferation index (Ki-67) …” Explicit mentioning of Ki-67 in context with proliferation index in the further (like line 220) is optional but can be left unchanged in the manuscript.
- Lines 119/120: Please rephrase to: "Although the prognostic significance of dual expressor status is still not entirely clear (…)"
- Lines 159/160 (PCDLBCL-LT): Bearring significance, please state also the absolute case number of the refferd lagerst series of patients.
- Lines 173/174 (PCFCL): To enhance readeability, please rephrase to: “On approximately 5% of cases the legs are involved. 15% of patients present multifocal skin lesions.”
- Lines 197/198: Please simplify by rephrasing as own sentence: “It must be differentiated (…)”
- Lines 262/263: Please rephrase this sentence to: "(…) lymphoma and diffuse large B-cell lymphomas occurring (…)"
1. Alaggio R, Amador C, Anagnostopoulos I, Attygalle AD, Araujo IBDO, Berti E, Bhagat G, Borges AM, Boyer D, Calaminici M et al: The 5th edition of the World Health Organization Classification of Haematolymphoid Tumours: Lymphoid Neoplasms. Leukemia 2022, 36(7):1720-1748.

Author Response
Reviewer #2:
In their manuscript “Primary cutaneous B-cell lymphomas with large cell morphology: a practical review.” Ronchi et al. review literature about rare primary b-cell lymphoma of the skin. Expanding upon the WHO-HAEM4R classification, the authors give interesting insight about the current molecular understanding of those entities and furthermore provide an argumentative baseline in favour or against defining PCBCL-NOS as separate entity.
Some formal/minor editing and aspects as well expansion upon methodical conception (see below) is needed before this review should be considered for publication.
Figures in the manuscript feature cincial findings and tissue samples. Please state if those are either original (unpuplished) data (if so, is the ethical statemanet truley not appicable) or are those cited from an other publication (if so, citations are, of course, mandatory).
AA: in the ethical statement, we stated that the figures are original/unpublished. The patients signed the informed consent to publish their clinical and histological images.
Et al. is usually abbreviated with a point. Please check this in the manuscript.
On several occasions a comma is used incorrectly before and. Please adjust this throughout the manuscript.
AA: thank you for your attention. We made the corrections.
Currently the manuscript is lacking important information about its methodology. The authors should elaborate on which merit articles were deemed suitable for inclusion into the review. Especially, was a (semi-)structured literature search used and which databases were included? Depending on the length of the methodology section this could either be presented as separate chapter or a paragraph at the end of the introduction.
AA: we added a paragraph at the end of the introduction.
- Line 19 (Abstract): The term "of the latter" is confusing in this context, mentioning secondary b-cell lymphoma of the skin just previously. I recommend not including it here and starting the sentence with: “A minority of cases (…)”
- Lines 25 & 59: Please replace "even overlapping (…)" with which are overlapping (…) or featuring overlapping to increase readability.
- Lines 34/35 (Introduction): Please rephrase here as two sentences, like: Most primary cutaneous lymphomas include small cells. T-cell lymphomas, deriving from mycosis fungoides, are the most common.
- Lines 48-50 (Introduction): Considering the upcoming WHO-HAEM5 which makes no alterations to PCBCL subtypes, I recommend outlining this and providing suitable citation [1]. Like: "Classification of PCBCLs with large cell morphology (which will remain unaltered in the upcoming edition of WHO classification) (…)"
- Lines 70 & 72 (PCDLBCL-LT): Occurrence in elderly women is stated somewhat redundantly. I suggest deleting it in the first sentence and rephrasing the latter one to: "Elderly Women are affected more often than males."
- Line 72 (PCDLBCL-LT): Please rephrase to: “Localization to head or neck are rare.”
- Line 73 (PCDLBCL-LT): Statement of the usual clinical presentation is used redundant. Please delete either "clinically" or "at the time of clinical observation" at the end of this sentence.
AA: we made the corrections.
Lines 75/76, 174/175, 272/273: Please elaborate upon the type of survival (overall, progression free, etc) if this is stated in the cited papers.
AA: we added data about type of survivals when available.
- Lines 76-79 (PCDLBCL-LT): Please provide suitable citation on prognostic factors of PCDLBCL-LT.
- Line 112 (PCDLBCL-LT): Germinal centre should be written out before abbreviating it the first time.
- Lines 115/116 (PCDLBCL-LT): Considering it is firstly stated in the text, please add: “Proliferation index (Ki-67) …” Explicit mentioning of Ki-67 in context with proliferation index in the further (like line 220) is optional but can be left unchanged in the manuscript.
- Lines 119/120: Please rephrase to: "Although the prognostic significance of dual expressor status is still not entirely clear (…)"
- Lines 159/160 (PCDLBCL-LT): Bearring significance, please also state the absolute case number of the refferd lagerst series of patients.
- Lines 173/174 (PCFCL): To enhance readeability, please rephrase to: “On approximately 5% of cases the legs are involved. 15% of patients present multifocal skin lesions.”
- Lines 197/198: Please simplify by rephrasing as own sentence: “It must be differentiated (…)”
- Lines 262/263: Please rephrase this sentence to: "(…) lymphoma and diffuse large B-cell lymphomas occurring (…)"
AA: we made the corrections.
